# Psychosocial hazard exposures and mental health outcomes among ambulance Emergency Medical Technicians in Ghana: A qualitative phenomenological study

Elias Kodjo Kekesi[1]*, Maxwell Asumeng[2], Ernest Darkwah[2], David Lackland Sam[3]

1 Department of Management Studies, Ghana Communication Technology University, Tesano-Accra, Ghana, 2 Department of Psychology, University of Ghana, Legon, Ghana, 3 Department of Psychosocial Science, University of Bergen, Bergen, Norway

* ekekesi@gctu.edu.gh

## Abstract

Emergency Medical Technicians (EMTs) face elevated risks of physical, mental, and psychosocial harm owing to the demanding nature of their work. Despite extensive research on the impact of emergencies on victims, the psychosocial hazard exposures and well-being of EMTs who respond first to these emergencies remain underexplored. We adopted an interpretative phenomenological approach to investigate psychosocial hazards and their associated mental health and work attitude outcomes among EMTs in Ghana's National Ambulance Service (NAS). In-depth interviews were conducted with 13 EMTs from seven of Ghana's 16 regions. Reflexive thematic analysis uncovered three core themes regarding the psychosocial hazards EMTs are often exposed to: "Bearing the Burden" which reflects the physical, emotional, and ethical strain of ambulance work; "Systemic Strain" which captures the institutional gaps and resource limitations that hinder effective emergency response; and "Between Stigma and Support", which encompasses the complex social dynamics and public perceptions that shape EMTs' professional experiences. These exposures resulted in negative outcomes, such as psycho-emotional distress (e.g., vicarious trauma, moral injury, fatigue, and depression), as well as positive outcomes, such as posttraumatic growth, resilience, and religious coping. Regarding the impact on work attitudes, participants reported lower job satisfaction and higher intention to quit. These findings have implications for enhancing EMT well-being and prehospital ambulance emergency care in Ghana and lend credence to the dual nature of psychosocial outcomes in high-risk work environments. To reduce these hazards and improve EMTs' well-being and resilience, psychosocial support should be integrated into organizational policies and EMT training.

**Data availability statement:** A minimal anonymized dataset containing de-identified participant quotes, demographic summaries, and thematic coding structures is provided as Supporting Information (S1 Table). Due to the sensitive nature of the qualitative interview data and the potential risk of participant identification, the full de-identified interview transcripts are not publicly available. The transcripts are securely archived in a restricted-access repository on the Open Science Framework (OSF) [https://osf.io/g84b9] to ensure long-term preservation. Requests for access may be directed to Dr. John Enoch Dotse, PhD, Senior Lecturer, Department of Psychology, University of Ghana (jekdotse@ug.edu.gh), who serves as a non-author institutional point of contact and was not involved in the study. Access will be granted for research purposes in accordance with the original ethical approval and conditions to protect participant confidentiality.

**Funding:** The author(s) received no specific funding for this work.

**Competing interests:** The authors have declared that no competing interests exist.

## Introduction

In healthcare systems worldwide, Emergency Medical Technicians (EMTs) are vital first responders who provide life-saving care in various situations, from mass casualty incidents and natural disasters to traffic accidents and cardiac arrest. In 2024 alone, global natural disasters claimed 16,753 lives, impacted over 167 million people, and caused US$241.95 billion in economic losses. This data highlights the increasing need for skilled emergency responders in diverse hazardous contexts [1, 2].

Ambulance EMTs provide prehospital care at accident or disaster scenes, in homes, and during ambulance transport. This group has been described as the least studied emergency medical service in Africa, despite its critical importance for initial patient stabilization and public safety [3–5]. Additionally, the nature of their work inherently exposes them to significant psychosocial hazards, including high work demands, traumatic event exposure, limited autonomy, and inadequate organizational support [6, 7].

Long-term exposure to these hazards has been shown to have negative physical, biological, social, and psychological effects on EMTs. Sleep disturbances, cardiovascular issues, musculoskeletal disorders, strained relationships, burnout, and an increased risk of posttraumatic stress disorder are common among EMTs [8–10]. Negative work-related impacts, such as lowered job satisfaction, employee engagement, and higher intention to quit, have also been reported [11–14]. Despite these negative outcomes, other studies have reported positive outcomes, with some individuals demonstrating resilience and undergoing posttraumatic growth, leading to stronger social ties and a greater sense of purpose [15–17].

Recent developments in Ghana, such as government initiatives for ambulance procurement and increased visibility during COVID-19, have raised public awareness of ambulance EMTs [18]. Nonetheless, EMTs in Ghana face additional psychosocial hazards due to operational issues such as ambulance breakdowns, delayed response times, inadequate healthcare infrastructure, and cultural stigma, where many Ghanaians associate ambulances primarily with death rather than life-saving care [19, 20]. We argue that Ghana's collectivist culture, characterized by communal responsibility and extended family involvement, may influence how EMTs cope with these hazards and seek support [21].

Despite the increasing exposure to psychosocial hazards, research on ambulance EMTs in Ghana and Africa remains limited. Although COVID-19 has spurred more research on healthcare workers, most of these studies have focused on in-hospital staff rather than prehospital EMTs, who face unique challenges at accident scenes, in homes, and during patient transportation [5, 22]. Similarly, Ghana's current mental health legislation focuses on the general population, with limited provisions for the specific psychosocial needs of emergency responders [23].

Using the Job Demands-Resources (JD-R) theory as a framework, which posits that employee well-being depends on the balance between job demands and available resources [24], this study was guided by the following three research questions:

1. What are the lived experiences of psychosocial hazards among EMTs working in Ghana's National Ambulance Service (NAS)?

2. What mental health outcomes and work attitudes result from exposure to psychosocial hazards?

3. How do cultural and resource constraints in Ghana's healthcare system influence EMTs' coping strategies and overall well-being?

The findings are intended to inform policies, interventions, and strategies aimed at improving the well-being and resilience of EMTs in Ghana's emergency response.

## Method

### Ethics statement

Ethical approval was obtained from the University of Ghana Ethical Committee for Humanities (protocol number: ECH 337/21–22). Informed consent was obtained through reading aloud the informed consent sheet to participants and their consent recorded verbally and signed on the sheet. Participants were made aware of their right to withdraw at any time, and coded identifiers were used to protect their confidentiality. A licensed clinical psychologist was available to provide support for psychological distress, although no participant required assistance.

### Study design and population

This phenomenological study employed semi-structured interviews to explore EMTs' subjective experiences of psychosocial hazards in Ghana's emergency medical services. The study was conducted across 12 ambulance service stations in seven of Ghana's 16 regions.

Ghana, with an estimated population of approximately 33 million people [25], achieved lower middle-income status in 2010 [26] but faces challenges in providing emergency medical services (EMS). The National Ambulance Service (NAS) and Ghana Health Service (GHS) oversee the prehospital ambulance emergency response system, which has approximately 3,500 EMTs and 356 operational ambulances spread across more than 260 stations by the end of 2024 [18]. EMTs, with at least one emergency response within the previous eight months and more than a year of field operational experience, were the inclusion criteria used to recruit participants. The eight-month period allowed for possible variations in call volume and short-term reassignments while guaranteeing recent operational exposure.

Official invitations to their stations and in-person visits were used to recruit EMTs. After 13 interviews, data saturation was reached. The sample consisted of 9 men and 4 women, aged 27–46 years, with 2–12 years of experience. Table 1 provides detailed demographic information on the study participants.

### Data collection procedures

A semi-structured interview guide (see S1 File) was developed based on existing literature and the Job Demands-Resources (JD-R) conceptualization. The guide was divided into three sections: (1) screening questions to confirm recent emergency response experience, (2) demographic data collection, and (3) exploration of psychosocial hazard experiences, coping strategies, and outcomes. Three emergency responders participated in a pilot study of the guide to ensure its dependability and applicability.

Depending on participants' availability and preference, the first author conducted eight telephone interviews and five in-person interviews at the participants' convenience. The interviews lasted between 40 and 60 minutes on average and were conducted over a three-week period, from July 22 to August 14, 2022. Each interview was audio-recorded and transcribed verbatim. Interview notes were kept to reduce bias and promote reflective analysis. Before each interview, rapport was built to promote candid sharing of experiences.

**Table 1. Personal interview demographic data.**

| Participant ID | Gender | Status/Position | Age (years) | Tenure (years) |
|---|---|---|---|---|
| EMT1 | Male | Basic EMT | 30 | 2 |
| EMT2 | Male | Senior Advanced EMT | 41 | 10 |
| EMT3 | Male | Basic EMT | 29 | 4 |
| EMT4 | Male | Senior EMT | 34 | 12 |
| EMT5 | Female | Senior EMT | 37 | 8 |
| EMT6 | Male | Advanced EMT (Station Manager) | 32 | 10 |
| EMT7 | Female | Senior EMT | 35 | 8 |
| EMT8 | Female | Senior EMT | 33 | 8 |
| EMT9 | Female | Basic EMT | 28 | 2 |
| EMT10 | Male | Principal Advanced EMT (Regional Manager) | 42 | 12 |
| EMT11 | Male | Principal Advanced EMT (Deputy Regional Manager) | 36 | 11 |
| EMT12 | Male | Advanced EMT | 43 | 10 |
| EMT13 | Male | Senior EMT | 41 | 7 |

## Data analysis

The six-phase reflexive thematic analysis (RTA) developed by Braun and Clarke [27] was used to analyze the data using both theoretical and inductive approaches to data analysis. Coding and theme organization were performed using the online qualitative analysis software Delve Tool. Transcript segments were used to create the initial codes, and themes and their subthemes were inferred from the data based on their salience.

To ensure validity, transcripts and themes were independently reviewed by the study supervisors (Authors 2, 3, and 4) through investigator triangulation. To ensure consistency and dependability, a final thematic framework was created by examining the data codes, main themes, subthemes, and their relationships.

## Methodological rigor

Trustworthiness was enhanced following Guba and Lincoln's [28] criteria through investigator triangulation, member checking, and confirmation of data saturation. Member checking was performed during the interviews to verify the participants' experiences, and three participants reviewed the final thematic framework. After ten interviews, data saturation was reached, and three more interviews confirmed the patterns that had been identified.

In-depth explanations of the phenomena and study context were used to address transferability [27]. An external audit by a qualitative researcher outside Ghana confirmed that the conclusions matched the data and theory. Confirmability was strengthened through audit trails documenting the study process and reflexive journal entries that captured interview reflections.

## Findings

The data revealed three main themes of psychosocial hazard exposure and three main themes of outcomes. Relevant quotes are labelled with 'EMT', followed by participant numbers. A summary of the main themes, subthemes and sample quotes is presented in S1 Table.

### Psychosocial hazard exposures

**Bearing the burden: The psychosocial strain in ambulance work.** The nature of ambulance work emerged as a significant source of psychosocial hazard exposure, encompassing *physical demands, risk of infection and injury, cognitive and emotional demands, work overload,* *moral injury,* and *inadequate work-life balance*.

**Physical demands:** EMTs emphasized the physical toll of their work, particularly the risks associated with lifting heavy patients, which often led to injuries, including spine-related diseases: "*Over time, we may develop spinal injuries from lifting patients; as pressure on the spine increases, we are likely to experience spinal problems such as back pain and waist pain. A colleague, for instance, was lifting a patient, and …it affected him a lot. At one point, he couldn't walk well without support; these are the types of hazards we face.*" *(*EMT3). Although not directly experienced by this participant, this observation reveals an embedded fear of the incident, highlighting the physically demanding nature of the role, which could have long-term health implications for the workers.

**Risk of Infection, Accident, or Death:** A recurring concern was the heightened risk of exposure to infections and injuries during emergencies, particularly when handling accident scenes involving fresh blood: "*And then infection risk—when you get to an accident scene and you are not careful, you are likely to get infected because you come in contact with fresh blood.*" (EMT2). Other hazards emerged from the physical environment, such as difficulties with poor road networks and possible run-ins with armed robbers during emergency operations. Participants emphasized the risks posed by potholes on roads, which not only endangered their safety but also caused collisions. The following environmental hazards put EMTs at risk for physical harm, deadly accidents, and psychological distress related to violent encounters or death: "*You know this our pothole road; it could make you hit any part of your body on a hardware…. Some of our colleagues have been involved in accidents, and a lady just recently died. In the night, too, armed robbers—when they hear our siren, they think we are police; they sometimes also attack us.* (EMT11).

**Cognitive and Emotional Workload:** Participants described how handling chaotic emergency scenes, frequently with limited resources, increases cognitive demands: "*Sometimes you even forget that you are an EMT... you get confused…*" (EMT 11). Managing chaotic scenes requires quick decision-making under pressure, which exacerbates stress levels and impacts task performance. Furthermore, witnessing traumatic events, including deaths, can place a heavy emotional burden on people. EMTs reported experiencing psychological distress and feelings of devastation: "*You become so much devastated and then you break down because you see lives lost, you become traumatized, especially when the case is very bad*, … *You've been able to manage the case at the scene. And then, upon arrival or at the triage unit, you lose the patient, and you become more traumatized.*" (EMT2).

**Work Overload:** EMTs described situations in which they were overloaded due to a backlog of cases and staff shortages. Many EMTs feel overburdened by the ever-present urgency of ambulance work and the need to balance multiple responsibilities. Burnout was exacerbated by a lack of recovery time: "*It's too much when the cases keep coming, because occasionally you can respond to three or four cases … you would become worn out and wish you had some time to rest.*" (EMT 3). One participant recounted how their station was manned by fewer personnel than required: "*…every ambulance station requires nine people to run three shifts, but here…four people are running it. So, at times one will go off and two will work for maybe 24 hours.*" (EMT 11). The burden became particularly acute for EMTs with administrative responsibilities, who must manage EMT operational duties alongside office work: *"…even as we are talking, I have to perform at least two or three roles at the office right now, after coming back from the scene... Now before I alone will sit down and write, I have to enter everything into the tablet. Mostly, sometimes I don't have adequate time to relax*" (EMT13).

**Moral Injury:** Participants also expressed feelings of frustration and self-doubt brought on by systemic resource limitations that prevented them from providing the best care possible: "*You feel like all your efforts have been in vain... you become more frustrated. Sometimes you would be thinking, maybe there is something you should have done, you didn't do, and you've lost the patient*" (EMT2). This quote captures a recurrent pattern of system-induced frustration as EMTs grapple with the moral conundrums of providing care under resource constraints, which frequently result in triage errors, exacerbating their frustrations and pushing them to consider the consequences of their choices: "*You wonder what you did wrong that made you lose the patient. What should you have done that could have made the patient survive*?" (EMT2).

**Inadequate work-life balance:** EMTs' inability to maintain a healthy work-life balance was evident in the interviews. The unpredictable nature of emergency calls often intrudes into personal time, contributing to emotional fatigue and

strained relationships. EMTs reported that emergency calls frequently disrupted planned family time, creating tension in their relationships. One participant explained how emergency calls could consume significant portions of personal time: "*...sometimes when you want to spend time with your family, a call can come in and take almost 3 hours to 8 hours of your time, and before you get back, your family is asleep*" (EMT11). This unpredictability made it difficult for EMTs to consistently attend family activities and obligations, contributing to feelings of role conflict. The physical and emotional exhaustion caused by ambulance work also hinders EMTs' ability to manage domestic responsibilities and sustain intimate relationships. Participants described how work-related fatigue and stress affected their capacity to engage in household duties and intimate relationships, often leading to misunderstandings with their spouses.

"*I come home, I am unable to sleep, and the next day... I won't be able to [help with household tasks] because I am tired. My wife won't understand, and she may think you are not helping her, and she will be complaining... You're still thinking about things, and when you are asked to help, you are not ready because there are a lot of things you are thinking about. When you go to bed, you are asked to engage in certain intimate activities, but you find it difficult and are unable to do so. In fact, it can affect your family life*" (EMT10).

The participant's description demonstrates how psychological preoccupation with work experiences combined with physical exhaustion creates a cascade of domestic challenges, including misunderstandings with spouses who may interpret their partners' withdrawal as a lack of commitment rather than work-related exhaustion. This indicates that psychosocial hazards extend beyond the workplace, impacting EMTs' ability to manage practical domestic responsibilities and maintain intimate marital relationships.

**Systemic strain: Navigating institutional gaps and resource scarcity.** Subthemes reflecting these systemic challenges included *unconducive employment conditions* and *logistical constraints* such as PPE shortages, "no bed syndrome," and understaffing. This main theme encapsulates the institutional gaps and resource shortages that impede effective ambulance response.

**Unconducive Employment Conditions:** Participants expressed dissatisfaction with their employment conditions, particularly regarding risk allowances, salary disparities, and promotion opportunities. These concerns demonstrate broader systemic issues within ambulance service organizations, where EMTs often face high-risk environments without adequate compensation or career advancement. A recurring concern was the absence of risk allowances despite the hazardous nature of the work. EMTs frequently operate under high-speed and high-stress conditions, exposing them to physical danger: "*You know the work we do is very dangerous, and with regard to transport, you see the speed at which we drive; anything can happen. Despite the dangers we face, we do not receive any risk allowance, unlike other health service workers. So, if there is any accident, there is no allowance for you or your family*" (EMT1). This quote alludes to the precariousness of ambulance emergency work and the lack of institutional safeguards for workers and their families in the event of an injury or death. The absence of risk-related compensation reflects the undervaluation of EMTs' contributions and contributes to a sense of vulnerability and neglect.

In addition to risk-related concerns, participants reported financial insecurity due to low salaries and limited promotional opportunities. These disparities were particularly stark when compared with those of other health professionals within Ghana's health system: "*Financially, I would say our salary is very bad. Imagine even me as a senior person; I'm at level 18, and I started 12 years ago … at level 16. I've gone on two promotions. When you have your qualification in other areas [health sectors], you qualify to be a director. Your starting point is even deputy director... your colleagues elsewhere are directors, and you are still at level 18 when they are at level 24 or 25... it's shameful the way we are treated*" (EMT10). This statement reflects the stagnation many EMTs experience in their careers, where years of service or academic qualifications do not necessarily translate into financial or professional growth or advancement. Such conditions foster feelings of inequity and discontent, eroding morale and reducing job satisfaction.

**Logistical Constraints:** Logistical constraints were found to be significant barriers to effective ambulance service delivery. Participants reported challenges related to *fuel shortages*, *inadequate infrastructure*, *shortages of medical supplies*, *hospital bed unavailability*, and *understaffing*, all of which hindered their ability to perform their duties efficiently and safely. One of the most frequently cited issues was fuel shortages, which directly impacted ambulance mobility and response time: "*Our ambulance can be out of commission [not motorable] due to a shortage of fuel… and you cannot move the ambulance*" (EMT5).This quote suggests that resource limitations can immobilize critical emergency services, delay care, and potentially endanger patient outcomes.

Participants also described inadequate office infrastructure, which negatively impacted both operational coordination and their professional identity: "*In many locations, we do not even have our own offices. So, we are actually perching with other institutions*" (EMT10). The lack of a dedicated workspace undermines organizational efficiency and reflects the broader systemic neglect of emergency medical services within the healthcare infrastructure.

Shortages of personal protective equipment (PPEs) and essential medical supplies were reported as safety concerns: "*Sometimes we experience shortages of gloves... we beg for gloves from the hospital when we go for referral cases, which is not the best*" (EMT1). They also recounted having to reuse single-use medical tools due to supply shortages: "*For the tools, you have to disinfect and use the same tools for the next patient, which is not good, but due to shortages, that is the only way to go*" (EMT1). These accounts highlight the risks faced by EMTs and patients when resource limitations compromise basic safety protocols.

A particularly distressing logistical challenge was the lack of hospital beds, commonly referred to as "no bed syndrome" in Ghana. EMTs described situations in which patients were denied admission because of bed shortages, sometimes resulting in death during transit: "*By the time we arrive there, they would tell us there is no bed... the patient is expired, dead in your ambulance*" (EMT10). This finding shows that systemic inefficiencies in hospital capacity can have fatal consequences and place emotional strain on EMTs who cannot secure timely patient care.

Beyond infrastructure and supply issues, participants identified *understaffing* as a significant systemic strain that increased their workloads and created additional professional risks. Insufficient personnel levels forced EMTs to work beyond scheduled hours and cover for absent colleagues, often with limited institutional support: "*We have three shifts... the last time one case came in while I was supposed to be off, but because my colleague was seriously sick, I had to be at work*" (EMT13). This participant further highlighted the professional vulnerability created by these situations: "*Assuming we are many, things of this nature wouldn't happen, and the sad thing is that when you're going on such unofficial trips and something happens, you will be held responsible for it. Meanwhile, you are just trying to help.*" (EMT13). These accounts reveal how understaffing creates a double burden for EMTs by increasing their workload while simultaneously placing them in professionally precarious positions where they assume additional responsibilities and risks without formal recognition or protection from the employer.

**Between stigma and support: EMTs in the eyes of society.** This main theme reflects the nuanced social interactions and public perceptions that shape EMTs' professional experience. Five subthemes emerged from the analysis: *insufficient organizational and public support*, *stigmatization and violence*, *poor coordination with receiving facilities*, *bystander disruptions*, and *negative public perceptions of the ambulance service*.

**Insufficient Organizational and Public Support***:* Participants described inadequate support from both their organization and the public, manifesting in several critical areas such as fuel allocation: "*The support we obtain most often is fuel support... but for one month everything is finished within a week, and you have about 3 weeks to the end of the month*" (EMT10). This resource shortage directly impacted service delivery and created tension with the public: "*The public is never ready to even provide us fuel support…. At times the public doesn't understand why they would call, and we would say there is no fuel.*" (EMT11). Participants highlighted a disconnect between government messaging and operational realities: "*The government is not supportive, so mostly the patient's relatives complain about why we are charging [fuel cost], while the government is telling the public that ambulance operation is free*" (EMT13). This misalignment creates

confusion among service users and places EMTs in difficult positions when interacting with patients and their families. EMTs noted the absence of psychological support systems within their organization: "*Unfortunately, as a service I'm yet to see any psychological unit that will 'psyche' us up after such chaotic scenes*" (EMT11). Participants described limited supervisory support, although some recognized systemic constraints affecting their supervisors: "*Supervisors' support, I must say, their support is very low, and I can't blame them that much because... they have a lot on their plate to deal with*" (EMT11).

**Relationship conflicts:** High-pressure environments led to interpersonal conflicts among EMTs. These conflicts typically emerged during critical incidents when team members were under extreme stress and resources were limited: "*Sometimes, we argue and shout at each other under the pressure of the [accident] scene, especially when resources are limited, or the driver isn't where they should be*" (EMT13); "*Sometimes we shout at each other when we get there. There is no seniority... everybody is under pressure... because of the limited resources, we are confused at the scene.*" (EMT11) These sometimes escalated: "*Sometimes when two of you are overwhelmed, you begin to shout at each other, which at times leads to fights and hatred. This has never happened to me... I got frustrated and scolded a junior colleague for not fixing oxygen well and she became cold to me. After 3 days I apologized.*" (EMT6).

**Stigmatization and Violence:** EMTs reported experiencing both stigmatization and violence stemming from public misconceptions about their roles and unrealistic expectations during emergencies. These negative interactions occurred across multiple contexts with different stakeholders. Participants faced physical aggression and hostility from patients and their relatives: "*When you pick patients who are very aggressive, they injure you… If you don't strap or tie them to their trolley, patients and relatives may quarrel or attack you if they feel you're not doing your job or are slow to respond.* (EMT2). Other healthcare colleagues within the broader healthcare system created tensions: "*On arrival, the nurses turn to harass us that we kept long, forgetting the distances and the nature of the road... they also think we don't have to rush them to take over the case. At times, these nurses do not value the services of the EMT... they think they are superior to EMTs.*" (EMT6). Social stigma stemmed from misconceptions about EMTs' role: "*A lot of people think that we carry corpses or dead bodies…when they ask you what work you do and you tell them you work with ambulance service, they begin to withdraw from you.*" (EMT6)

**Poor coordination with the receiving hospital:** Participants recounted instances of delays, disagreements, and outright rejection of patients by hospital staff. One EMT described how, upon arriving at a receiving facility with a patient in the ambulance, hospital staff sometimes claimed that they had not received pre-arrival notification, often because of poor internal documentation. This left EMTs to wait or, in some cases, led to the patient being rejected altogether: "*Sometimes when we get to the receiving facility, they say they did not receive any call...you have to wait for them, or they reject the case.*" (EMT12). This lack of coordination not only delays critical care but also creates distress for EMTs, patients, and their relatives. These coordination lapses reveal systemic inefficiency. In some cases, EMTs arrive at hospitals only to find that there are no available beds or that the staff is unprepared to receive the patient, resulting in delays in care. Such situations often prolong EMTs' exposure to distress and, in severe cases, lead to patient death en route: "*Even the hospital staff who should understand the system—but no, their back-and-forth and arguments with drivers and us delay victims for hours, and at times, they would force us to keep the patients in our ambulance, and occasionally the patient dies in the process.*" (EMT11)

**Public Interference and Misperception:** EMTs reported challenges stemming from problematic public interactions, ranging from direct interference with ambulance care to broader negative perceptions of their profession. Untrained bystanders frequently disrupted ambulance response efforts by taking premature action at accident scenes: "*By the time you reach there, the bystanders would have taken the victims away.*" (EMT1). Although often well-intentioned, these interventions compromise patient safety and create additional challenges for EMTs attempting to provide appropriate care in chaotic environments. Beyond direct interference, participants described a pervasive lack of public understanding and appreciation for their role: "*I don't think the public recognizes us for what we do… only a few people appreciate our efforts.*" (EMT6). This lack of recognition diminishes the societal value of ambulance service in Ghana.

## Outcomes of psychosocial hazards exposures

The findings illustrate the important consequences of exposure to psychosocial hazards. The findings included three main themes: *negative psychosocial outcomes* (i.e., psycho-emotional distress and maladaptive coping), *positive outcomes,* and *work attitudes*.

**Negative psychosocial outcome: Psycho-emotional distress.** Psychosocial hazard exposure resulted in several negative outcomes for EMTs, particularly in terms of psycho-emotional distress (i.e., *vicarious trauma*, *work rumination*, *moral injury, mental exhaustion,* and *depression*) and maladaptive coping strategies.

**Vicarious Trauma***:* EMTs experienced trauma from repeated exposure to harrowing scenes, reflecting symptoms of vicarious trauma, that is, the emotional residue left from witnessing others' suffering and death, often over prolonged periods: *"At times, I experienced nightmares where I could see the body burned and other distressing scenes. And you get traumatized"* (EMT11). This recollection illustrates how graphic and violent scenes can intrude into one's personal life, disrupting sleep and emotional stability. *"You become confused. Victims shouting, pain everywhere, human parts scattered, blood everywhere. That makes you confused. And when you get back, psychologically you are not yourself. You get nightmares, can't sleep or eat"* (EMT6).

**Work Rumination:** Participants reported that their emotionally charged experiences lingered beyond shifts: *"When you lose a patient...even when you return home, you can still think about it...you think about it for…days"* (EMT2). Such reflections point to how unresolved emotional experiences from the field can intrude into personal time, potentially affecting sleep, relationships, and overall well-being.

**Mental Exhaustion**. Participants described physical exhaustion that intertwined with mental fatigue: *"You know the body is also a machine, and when you overwork, it gets tired. When you work, at a point you get too stressed out and even forget what you are supposed to do. Even after coming back from attending to cases, you still have documentation to do."* (EMT13). This suggests that the fatigue experienced was not merely physical but mental, characterized by diminished cognitive functioning due to stress and overexertion. Forgetting tasks or procedures in high-stakes environments may indicate compromised executive functioning, which may stem from chronic exposure to intense workloads without adequate recovery time.

**Depression**: Depression emerged with cognitive and somatic symptoms: "*I sometimes become confused, emotionally disturbed, and depressed at such scenes, even to the point that it affects my thinking, feeling, and ability to move around work successfully. After one of such trips*, I felt sick, but after some medical tests, everything turned out negative; it was more psychological—it was depression"* (EMT6). This narrative illustrates how depression in this context may not always present with overt sadness but rather with cognitive and somatic symptoms such as confusion, emotional numbness, and physical fatigue, which interfere with daily functioning.

**Maladaptive coping:** Some EMTs resorted to unhealthy coping mechanisms, particularly alcohol use, to manage stress and trauma. A regional manager acknowledged the prevalence of alcohol dependency among subordinates: "*We have people who are addicted to alcohol. Yes, and some of them are unable to stop. We advise them, we talk to them. … There was a time I had three serious drunkards at one station, and the CEO [Executive Head of NAS] came and was like, "Why have you put all these guys here? You need to separate them."* (EMT 10). Another participant described the personal use of alcohol to manage the psychological impact of traumatic scenes: "*Because of time delays, a victim can pass on in your care. Some of the scenes we witness are terrible and horrible, so at times I have to take in alcohol to even have an appetite for food.*" (EMT 13)

**Positive outcomes: Growth and coping.** Despite these challenges, EMTs have demonstrated resilience and positive growth. Posttraumatic growth and adaptive coping, such as resilient and religious coping, have also been reported.

**Post-traumatic Growth:** EMTs reported positive transformative experiences stemming from their exposure to hazards: *"It has changed my personal life in a positive way. I relate freely with everyone, regardless of my position"* (EMT11); *"I appreciate life more and understand the value of life"* (EMT2). These narratives resonate with the concept of posttraumatic growth, wherein individuals find meaning and personal development through adversity.

**Adaptive coping:** EMTs employed adaptive strategies, including *resilient coping,* when mental preparedness is often used: *"I have psyched myself already that these things are bound to happen...psychologically, it won't affect me that much."* (EMT3), and *religious coping*, with participants citing prayer and worship songs as sources of comfort: "*If I'm psychologically down with those problems, what I do to help myself is, I have some worship [religious] songs on my phone, that I believe once I'm listening to them, my problems go away."* (EMT4)

### Work attitudes

The findings revealed that EMTs' work attitudes, such as *job satisfaction* and *turnover intention*, were shaped by intrinsic motivation, such as the fulfillment derived from saving lives, and extrinsic factors, including remuneration, career progression, and organizational support.

**Low job satisfaction**: Participants expressed mixed feelings. While they valued the life-saving nature of their work, dissatisfaction with compensation and working conditions tempered their overall job satisfaction: *"For the job itself, I am satisfied...but the nature of the work and the salary, I am not satisfied"* (EMT5). This quote illustrates the duality of EMTs' experiences, where the intrinsic reward of contributing to public health is undermined by inadequate financial and systemic support for their work. Low remuneration emerged as a dominant concern, with participants linking financial insecurity to a diminished morale: *"If your pocket is not okay, it won't make you happy [satisfied]."* (EMT10)

**Intentions to quit:** Dissatisfaction with limited career advancement opportunities, which contributed to feelings of stagnation and undervaluation, translated into high quit intentions. EMTs expressed a willingness to leave the NAS if better financial opportunities emerged elsewhere: *"If I have any opportunity, I will not even look at what I am going to do; I will look at how much I am going to get. … And if that place is better, I will leave"* (EMT10); "*I have thought of leaving given a better opportunity, because I think I have been with the service for a long time, and the level I should be is not where I am*." (EMT 6). Such sentiments reflect systemic organizational challenges that threaten workforce stability.

## Discussion and conclusion

### Psychosocial hazard exposures in ambulance work

This study explored the psychosocial hazard exposures and outcomes experienced by EMTs in Ghana, revealing three interconnected domains of hazards that systematically undermine their well-being and effectiveness. These domains captured in the themes 'Bearing the Burden', 'Systemic Strain,' and 'Between Stigma and Support' reveal how individual, organizational, and societal factors converge to create a challenging work environment for ambulance emergency responders in low-resource settings.

**The multifaceted nature of psychosocial hazards.** The theme "Bearing the Burden" reveals how the inherent demands of ambulance work create cascading physical, cognitive, and emotional consequences. EMTs described musculoskeletal strain from patient lifting, constant infection risks, and cognitive overload from managing chaotic emergency scenes with limited resources. The cognitive overload described by the participants is a well-documented precursor of emotional exhaustion and impaired decision-making in high-pressure settings [29]. Particularly striking was the prevalence of moral injury, described as the psychological damage that occurs when individuals are unable to act according to their moral beliefs due to systemic constraints [30]. EMTs frequently describe feeling helpless when patients die due to resource limitations or system failures, leading to persistent self-doubt and questioning of their professional competence. These findings align with the JD-R model, where excessive job demands deplete employees' psychological and physical resources, leading to burnout and health problems [31]. This finding extends beyond typical occupational stress research to highlight how resource scarcity in low-income settings creates unique ethical dilemmas that compound traditional job stress.

**Systemic failures of chronic stressors.** Chronic understaffing, inadequate compensation, fuel shortages, and equipment deficits created a work environment in which EMTs felt systematically undervalued and unsupported. These

findings mirror research from other low-resource healthcare settings, where systemic underfunding creates cascading effects on worker well-being [32–34].

Participants' descriptions of financial insecurity went beyond job dissatisfaction to include a basic psychosocial hazard that led to chronic stress that extended into EMTs' personal lives. Insufficient resources exacerbate these systemic problems, which not only impede emergency response but also cause serious moral harm and ethical quandaries for EMTs. This was especially evident during the COVID-19 pandemic, when the lack of PPEs was associated with increased anxiety and burnout among frontline healthcare workers [22].

Addressing these systemic resource constraints will require innovative yet contextually appropriate approaches to resource tracking and allocation. Healthcare supply chain management in developing countries faces fundamental challenges including inadequate infrastructure, limited technical capacity, and governance weaknesses that result in the fuel shortages, equipment deficits, and supply chain disruptions evident in this study [35]. In the context of African emergency medical services specifically, strengthening resource management systems is critical not only for improving service delivery but also for reducing the occupational stressors experienced by frontline providers.

Digital resource tracking systems offer promising avenues for improving transparency and accountability in the NAS. Mobile health (mHealth) solutions have demonstrated potential for addressing supply chain challenges in African healthcare contexts, though successful implementation requires careful attention to local infrastructure, user capacity, and sustainability considerations [34, 36]. For Ghana's NAS, phased implementation of mobile-based tracking systems for fuel distribution, equipment maintenance, and supply chain management could provide real-time visibility into resource utilization while remaining feasible within existing technical capacity. Other advanced machine learning models, such as ensemble learning [37], can accurately classify transactional data and could be used to monitor and secure resource allocation but their implementation may not be practical in Ghana yet.

**Social dynamics and professional isolation.** EMTs' everyday experiences are shaped by broader social healthcare system dynamics, as the theme 'Between Stigma and Support' revealed. Public misconceptions about the ambulance service, such as the persistent belief that ambulances merely transport corpses, have created a social stigma that has isolated EMTs from community support. Poor coordination with receiving hospitals, particularly the 'no-bed syndrome,' exacerbated this stigmatization, resulting in treatment delays and additional moral distress. This finding is consistent with research indicating that EMS personnel are frequently exposed to aggression and violence, which are significant predictors of psychological distress [38].

These coordination failures represent systemic breakdowns that extend beyond individual hospital policies to reflect broader inadequacies in the healthcare system. Poor ambulance–hospital coordination is a widely documented barrier to effective patient care [39]. When EMTs described patients dying in ambulances while waiting for hospital admission, this highlighted how inter-organizational failures create profound psychological trauma for frontline workers who witness preventable deaths.

## Psychosocial outcomes and adaptive responses

Participants reported vicarious trauma from repeated exposure to human suffering, persistent work rumination that intruded into personal time, and depressive symptoms that impaired their professional and personal functioning. These outcomes align with research identifying emergency responders as high-risk populations for PTSD, depression, and burnout [40]. These findings further corroborate the JD-R model, indicating that trauma exposure and work overload deplete psychological resources and lead to negative health outcomes [31].

The prevalence of work rumination, the inability to mentally detach from work stressors, has emerged as particularly concerning and is linked to emotional exhaustion [41]. The absence of structured mental health interventions within Ghana's emergency services leaves EMTs relying on informal or maladaptive coping mechanisms, which is particularly

concerning given the pervasive mental health stigma in Ghana, where psychological distress is often misinterpreted through spiritual lenses or viewed as a personal weakness [21].

However, our findings also revealed important adaptive responses that complicate the simple narratives of occupational harm. Some EMTs reported posttraumatic growth, describing how their work enhanced their appreciation of life and strengthened their interpersonal relationships. This finding suggests that Ghana's collectivist cultural context may provide unique resources for creating meaning and resilience. In Ghana's collectivist society, strong community bonds, family support, and shared spiritual beliefs provide a powerful framework for meaning-making and coping with the aftermath of trauma [42].

The prevalence of maladaptive coping strategies, particularly alcohol use, highlights the lack of formal support systems in Ghana's emergency services. When EMTs describe drinking to 'forget' traumatic scenes or to stimulate their appetite after difficult calls, this reveals how the lack of structured mental health interventions forces workers toward potentially harmful self-medication strategies. This gap is particularly concerning, given the absence of psychological first aid programs recommended by Tessier et al. [5] for Ghanaian emergency service personnel.

**Cultural context and resilience resources.** Ghana's cultural context creates both challenges and resources for EMTs' psychological well-being. While mental health stigma prevented many from seeking formal support, strong family bonds, community connections, and spiritual beliefs provided alternative frameworks for coping with occupational stress. This duality, in which cultural factors serve as both barriers and resources, distinguishes the Ghanaian experience from Western contexts and emphasizes the importance of culturally adapted interventions.

Religious coping emerged as particularly significant, with EMTs describing how spiritual practices helped them process traumatic experiences and maintain hope despite the systemic challenges. This finding aligns with research on African contexts, where spiritual frameworks often provide primary resources for psychological healing [42].

## Work Attitudes and career sustainability

The tension between intrinsic motivation and extrinsic frustration profoundly shapes EMTs' work attitudes and career intentions. While participants derived deep satisfaction from saving lives and serving their communities, this was consistently undermined by financial insecurity, limited career advancement opportunities, and feelings of being undervalued by both their organization and society. These findings are consistent with research emphasizing the critical roles of fair compensation and professional development in retaining emergency service personnel [43].

From a JD-R perspective, the life-saving nature of ambulance work represents a powerful job resource that can buffer occupational stressors. However, our findings suggest that when the basic needs for fair compensation and professional recognition remain unmet, even the most meaningful work becomes insufficient to maintain job satisfaction and reduce turnover intention. Within Ghana's economic context, where cost-of-living pressures are acute, inadequate remuneration creates chronic financial stress that compounds other psychosocial hazards [18].

The high turnover intentions reported by participants represent more than individual career decisions; they threaten the sustainability of Ghana's emergency medical system and the quality of care available to the population. When experienced EMTs leave due to poor working conditions, it creates a cycle in which the remaining staff face increased workloads and stress, potentially accelerating further quits.

## Implications for theory and practice

The findings extend occupational health theory by demonstrating how psychosocial hazards manifest differently across different cultural and economic contexts. While the JD-R model provides a useful framework for understanding EMT experiences, our results suggest that traditional Western conceptualizations of job demands and resources may not fully capture the complexity of working conditions in low-resource settings in non-Western countries. The systemic nature of hazards faced by Ghanaian EMTs, from fuel shortages to hospital coordination failures, highlights how individual-level interventions alone are insufficient to address occupational health challenges in resource-constrained environments.

Effective interventions must simultaneously address the organizational, healthcare system, and societal factors that shape EMTs' work experiences.

Future strategies for managing the NAS's limited resources and preventing financial insecurity should explore secure and transparent data systems. Advanced machine learning models, such as ensemble learning, can accurately classify transactional data and could be used to monitor and secure resource allocation [37]. Implementing blockchain-based resource tracking systems could provide real-time visibility into fuel distribution, equipment maintenance, and supply chain management, thereby reducing opportunities for mismanagement and ensuring that resources reach frontline EMTs. Such technological innovations, adapted to Ghana's infrastructure and capacity, could transform resource governance in the NAS and similar emergency services across LMICs. However, successful implementation would require initial investment in digital infrastructure, technical training for administrators and EMTs, and sustained institutional commitment to transparency and accountability. Pilot programs in high-volume urban stations like Accra and Kumasi could provide valuable lessons for national scale-up, with adaptations necessary for rural contexts where connectivity and technical capacity may be more limited [36]. Critically, technological solutions must be accompanied by institutional commitment to transparency, accountability, and data-driven decision-making to address the governance challenges underlying current resource management failures.

Our findings also contribute to the growing recognition that occupational trauma can produce both harmful and growth-promoting outcomes. The posttraumatic growth reported by some participants suggests that culturally grounded interventions might leverage indigenous resilience resources while addressing systemic stressors.

## Recommendations for policy and practice

Addressing psychosocial hazards requires coordinated multi-level intervention. Organizational reforms should prioritize fair compensation through formal salary reviews by the Ministry of Health and Fair Wages and Salaries Commission, establish clear career advancement pathways within the NAS, and strengthen supply chain management with mobile-based tracking systems adapted to Ghana's infrastructure to ensure consistent availability of vehicles, fuel, equipment, and supplies. Implementation requires NAS leadership engagement with relevant ministries and EMT representatives, beginning with phased introduction in high-volume urban stations followed by periodic reviews to ensure equity and sustainability across all regions.

Healthcare system coordination must address the 'no-bed syndrome' through mandatory patient handover protocols and real-time bed management systems. Key components include standardized checklists adapted to Ghana's context capturing essential patient information, joint EMT-hospital staff training emphasizing patient safety and conflict resolution, designated handover zones with appropriate staffing during peak hours, clear escalation procedures with designated contacts and maximum waiting times, and integration of handover metrics into performance evaluations for both EMTs and hospitals creating mutual accountability. Pilot implementation in urban stations should inform national scale-up with rural adaptations, requiring commitment from both NAS leadership and hospital administrators supported by ongoing monitoring.

Mental health support systems should include confidential Employee Assistance Programs staffed by professionals trained in first responder mental health, alongside structured peer support programs adapting proven models while leveraging Ghana's collectivist cultural values [44]. Implementation requires investment in professional mental health services, training for peer support coordinators, and integration of mental health resources into NAS organizational structures, with consideration for geographic accessibility of services across urban and rural areas and stigma reduction efforts to encourage help-seeking behavior among EMTs facing cultural barriers to mental health support.

Training curricula should incorporate modules on stress management, resilience building, and trauma-informed care adapted to local contexts. Effective implementation requires collaboration between NAS leadership and mental health professional associations like the Ghana Psychological Association to develop culturally appropriate training materials,

integration of mental health training into existing EMT certification and continuing education programs, and ongoing evaluation of training effectiveness through participant feedback and assessment of changes in EMT well-being and job satisfaction, with training delivered through accessible formats including in-person sessions, digital modules where connectivity permits, and regular refresher courses.

Public education initiatives must address misconceptions in the ambulance service and promote community support for EMTs. The NAS, in partnership with the National Commission for Civic Education, should launch sustained awareness campaigns while building formal relationships with the Ghana Police Service to ensure scene safety and with community leaders to promote respect for emergency responders. Implementation requires strategic planning for campaign delivery across multiple media channels including radio, television, and social media platforms, engagement with religious and traditional leaders to reach diverse communities, and establishment of community liaison roles within NAS to maintain ongoing dialogue with the public and address concerns.

### Limitations and future research

Some limitations must be considered when interpreting the findings of this study. As a qualitative study conducted in Ghana, the findings may not be generalizable to EMTs in other African countries or elsewhere, where working conditions, cultural contexts, or organizational structures differ substantially. Our sample may have excluded EMTs who were unwilling or unable to participate because of work demands or personal circumstances, potentially limiting the representation of the most severely affected EMTs. Additionally, the sensitive nature of mental health topics in Ghana's cultural context may have influenced participants' willingness to fully disclose their psychological distress.

Future research should employ longitudinal designs to track the development of psychological outcomes over time and evaluate the effectiveness of the intervention strategies. Mixed-methods approaches can quantify the prevalence of specific hazards and outcomes across larger more diverse samples of EMTs.

## Conclusion

Our findings demonstrate that EMTs' well-being is compromised not only by the inherent trauma of ambulance emergency response but also by deep-rooted organizational failures and challenging socio-professional environments. The resulting outcomes, including vicarious trauma, emotional exhaustion, moral injury, and high turnover intention, threaten both the sustainability of the EMT workforce and the quality of emergency care. This study highlights the need for a coordinated, multilevel strategy that moves beyond generic solutions to implement evidence-based interventions tailored to Ghana's unique cultural, economic, and systemic landscape.

Our findings also highlight important cultural resources for resilience and adaptation that can inform intervention development. The posttraumatic growth and spiritual coping strategies reported by some participants suggest that culturally grounded approaches might effectively support EMT well-being while addressing systemic challenges.

These findings further emphasize the recognition that EMT well-being cannot be separated from broader healthcare system functioning and societal support. Thus, effective interventions must simultaneously address individual, organizational, and systemic issues while leveraging cultural and community resources. By implementing comprehensive, culturally adapted strategies that move beyond generic solutions to evidence-based interventions tailored to Ghana's unique landscape, stakeholders can create more supportive work environments that protect EMT well-being while ensuring high-quality ambulance emergency care.

### Generative AI tools usage

We used ChatGPT as a proofreading tool to enhance the clarity and readability of our writing. The ideas, arguments, and structure of the paper were entirely developed by us. ChatGPT, Quillbot and Paperpal were used to refine the wording, grammar and improve coherence without altering the substance of our paper.

## Supporting information

**S1 Table. Thematic framework for psychosocial hazard exposures in Ghana's National Ambulance Service.**
(PDF)

**S1 File. Interview Guide.**
(PDF)

**S1 Checklist. Inclusivity in global research.**
(DOCX)

## Acknowledgments

This paper is part of the doctoral research conducted by the first author under the supervision of the other named authors. The first author was a beneficiary of the University of Ghana-Carnegie Corporation of New York's BANGA-Africa Doctoral Scholarship and is grateful for supporting the fieldwork component of this research. We thank the Emergency Medical Technicians and staff of the National Ambulance Service of Ghana for their participation and invaluable insights that made this study possible.

## Author contributions

**Conceptualization:** Elias Kodjo Kekesi, Maxwell Asumeng, Ernest Darkwah, David Lackland Sam.

**Data curation:** Elias Kodjo Kekesi.

**Formal analysis:** Elias Kodjo Kekesi.

**Investigation:** Elias Kodjo Kekesi.

**Methodology:** Elias Kodjo Kekesi.

**Project administration:** Elias Kodjo Kekesi.

**Supervision:** Maxwell Asumeng, Ernest Darkwah, David Lackland Sam.

**Validation:** Maxwell Asumeng, Ernest Darkwah, David Lackland Sam.

**Writing – original draft:** Elias Kodjo Kekesi.

**Writing – review & editing:** Elias Kodjo Kekesi, Maxwell Asumeng, Ernest Darkwah, David Lackland Sam.

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
