## [Decision Letter · Decision Letter 0]

14 Jan 2026

PMEN-D-25-00415

Psychosocial Hazard Exposures and Mental Health Outcomes Among Ambulance Emergency Medical Technicians in Ghana: A Qualitative Phenomenological Study

PLOS Mental Health

Dear Dr. Kekesi,

Thank you for submitting your manuscript to PLOS Mental Health. After careful consideration, we feel that it has merit but does not fully meet PLOS Mental Health’s publication criteria as it currently stands. Therefore, we invite you to submit a revised version of the manuscript that addresses the points raised during the review process.

The manuscript required revision to enhance its clarity and impact. Key revision include aligning the abstract with full findings, reframing the study aims as clear research questions. Also, please elaborate on the practical implementation of key policy recommendation and consider referencing relevant technological frameworks such as secure data classification models, for managing resource constraints.

We look forward to receiving your revised manuscript.

Kind regards,

Hanif Abdul Rahman, Ph.D.

Academic Editor

PLOS Mental Health

Journal Requirements:

1. Please include a complete copy of PLOS’ questionnaire on inclusivity in global research in your revised manuscript. Our policy for research in this area aims to improve transparency in the reporting of research performed outside of researchers’ own country or community. The policy applies to researchers who have travelled to a different country to conduct research, research with Indigenous populations or their lands, and research on cultural artefacts. The questionnaire can also be requested at the journal’s discretion for any other submissions, even if these conditions are not met.  Please find more information on the policy and a link to download a blank copy of the questionnaire here: https://journals.plos.org/mentalhealth/s/best-practices-in-research-reporting. Please upload a completed version of your questionnaire as Supporting Information when you resubmit your manuscript.

Additional Editor Comments (if provided):

Reviewers' comments:

Reviewer's Responses to Questions

**Comments to the Author**

1. Does this manuscript meet PLOS Mental Health’s publication criteria ? Is the manuscript technically sound, and do the data support the conclusions? The manuscript must describe methodologically and ethically rigorous research with conclusions that are appropriately drawn based on the data presented.

Reviewer #1: Yes

Reviewer #2: Yes

Reviewer #3: Yes

2. Has the statistical analysis been performed appropriately and rigorously?

Reviewer #1: Yes

Reviewer #2: N/A

Reviewer #3: N/A

3. Have the authors made all data underlying the findings in their manuscript fully available (please refer to the Data Availability Statement at the start of the manuscript PDF file)?

Reviewer #1: Yes

Reviewer #2: Yes

Reviewer #3: Yes

4. Is the manuscript presented in an intelligible fashion and written in standard English?

Reviewer #1: Yes

Reviewer #2: Yes

Reviewer #3: Yes

Reviewer #1: 1. Ensure that the main outcomes in the abstract match those in the discussion section. The abstract mentions "vicarious trauma, fatigue, and depression" as negative outcomes but does not include moral injury, which is a key finding discussed later. Consider adding 'moral injury' to the abstract to highlight this important finding.

2. Rephrase the study aims into two or three clear research questions. For example: What are the lived experiences of psychosocial hazards for EMTs in Ghana's NAS? What mental health issues and work attitudes result from these experiences? How do cultural and resource constraints in Ghana affect EMTs' coping and well-being?

3. While the abstract and introduction summarize the study, the three specific aims in the Introduction (lines 387–391) are quite detailed. These should be framed more simply as clear research questions to guide the reader.

4. The current data availability statement is weak and may not meet PLOS journal requirements, which usually require full public access to a minimal dataset upon acceptance. Simply stating that raw transcripts will be available upon request is often not sufficient for open data policies.

5. To strengthen the discussion on systemic failures and future healthcare data management, especially regarding potential fraud and decision-making in the National Ambulance Service (NAS), the authors should reference reliable advanced data processing research. A paper on ensemble machine learning for classifying blockchain transactional data is relevant since blockchain is mentioned in the study and could be a solution for transparent resource tracking.

6. In the Discussion under "Systemic failures of chronic stressors" (or as a new paragraph in the Conclusion), add the following: "Future strategies for managing the NAS's limited resources and preventing financial insecurity should explore secure and transparent data systems. Advanced machine learning models, such as ensemble learning, can accurately classify transactional data and could be used to monitor and secure resource allocation. It is suggested for the authors to refer the paper: “An Ensemble Machine Learning-Based Model for Blockchain Transactional Data Classification”.

7. The policy recommendations presented are good, but some, like "mandatory patient handover protocols," need more details on how they would be implemented in Ghana's National Ambulance Service (NAS) context to show their practicality.

Reviewer #2: This was a very interesting and informative piece of research. Research on this topic is understudied, especially in West African countries. The research ended with good, practical recommendations for implementation.

Reviewer #3: This manuscript presents a well- executed qualitative phenomenological study examining psychosocial hazard exposures and mental health outcomes among ambulance emergency medical technicians (EMTs) in Ghana. It addresses a critical and underexplored area within global and occupational mental health, particularly in low- and middle-income country contexts, and aligns strongly with the aims and scope of PLOS Mental Health. The study design is well justified. The manuscript is well written, with clear structure and intelligent English. Overall, this is a high-quality and valuable contribution to the literature on emergency medical services and occupational mental health in resource-constrained settings. I have no substantive concerns and recommend acceptance of the manuscript as submitted.

**Do you want your identity to be public for this peer review?** For information about this choice, including consent withdrawal, please see our Privacy Policy .

Reviewer #1: No

Reviewer #2: **Yes:** Aylana BrewsterAylana Brewster

Reviewer #3: **Yes:** SIDRA BATOOLSIDRA BATOOL

---

## [Decision Letter · Decision Letter 1]

11 Mar 2026

Psychosocial Hazard Exposures and Mental Health Outcomes Among Ambulance Emergency Medical Technicians in Ghana: A Qualitative Phenomenological Study

PMEN-D-25-00415R1

Dear Dr Kekesi,

We are pleased to inform you that your manuscript 'Psychosocial Hazard Exposures and Mental Health Outcomes Among Ambulance Emergency Medical Technicians in Ghana: A Qualitative Phenomenological Study' has been provisionally accepted for publication in PLOS Mental Health.

Best regards,

Hanif Abdul Rahman, Ph.D.

Academic Editor

PLOS Mental Health

Reviewer Comments (if any, and for reference):

Reviewer's Responses to Questions

**Comments to the Author**

Reviewer #1: All comments have been addressed

Reviewer #2: All comments have been addressed

publication criteria ? Is the manuscript technically sound, and do the data support the conclusions? The manuscript must describe methodologically and ethically rigorous research with conclusions that are appropriately drawn based on the data presented.

Reviewer #1: Yes

Reviewer #2: Yes

3. Has the statistical analysis been performed appropriately and rigorously?

Reviewer #1: Yes

Reviewer #2: N/A

4. Have the authors made all data underlying the findings in their manuscript fully available (please refer to the Data Availability Statement at the start of the manuscript PDF file)?

Reviewer #1: Yes

Reviewer #2: Yes

5. Is the manuscript presented in an intelligible fashion and written in standard English?

Reviewer #1: Yes

Reviewer #2: Yes

**Reviewer #1:**  I recommend accepting this revised manuscript in the journal. The authors have shown a strong commitment to the peer-review process by addressing all previous concerns. They have resolved each reviewer comment from the initial round with thoughtful changes to the text and structure. The final version of the manuscript is now strong, well-presented, and ready for the scientific community. I recommend accepting this revised manuscript in the journal. The authors have shown a strong commitment to the peer-review process by addressing all previous concerns. They have resolved each reviewer comment from the initial round with thoughtful changes to the text and structure. The final version of the manuscript is now strong, well-presented, and ready for the scientific community.

1. The authors responded comprehensively to reviewer suggestions, addressing research questions, data access, and technological solutions.

2. They improved clarity by clearly defining three research questions.

3. The study uses a solid interpretative phenomenological approach and reflexive thematic analysis suitable for qualitative research.

4. It offers a novel, important contribution to global mental health by exploring overlooked psychosocial aspects of prehospital emergency technicians in West Africa.

5. The analysis is strengthened by including "moral injury" as a key theme, enhancing understanding of negative effects from resource shortages.

6. The manuscript's quality improved through careful proofreading, detailed policy suggestions, and a better Data Availability Statement.

**Reviewer #2:**  Additions and amendments are well written Additions and amendments are well written

**Do you want your identity to be public for this peer review?** For information about this choice, including consent withdrawal, please see our Privacy Policy .

Reviewer #1: No

Reviewer #2: No
